# Video Summarization Pretraining with Self-Discovery of Informative Frames

## Abstract

The rapid proliferation of videos makes automated video summarization (VS) an essential research problem: "Which abridged video best conveys the whole story?" The limited size of datasets is known to constrain the generalization of advanced VS methods, requiring advanced pretraining techniques to capitalize on unlabeled videos. Several pretraining methods for VS have been proposed. Yet, they heavily rely on fixed pseudo-summaries, often fail to capture the diverse frame importance, resulting in narrow generalization. To resolve conflicts between pseudo-summaries and downstream tasks, our idea is: First, pretraining should enable the summarizer to learn how to distinguish more meaningful summaries from unlabeled videos, without perspective differentiation; In this way, finetuning only requires adapting the pretrained multifaceted importance to the downstream perspective, facilitating supervised learning. Our pretraining approach, named ViSP, is free of pseudo-summaries, expecting to better align with the ill-posed nature of defining keyframes. The pre-trained model can be fine-tuned to create the SOTA summarizers by leveraging the knowledge base behind frame saliency. ViSP is conceptually simple and empirically powerful, and it can be used to pre-train any neural video summarizer. Extensive experiments on two benchmark datasets (SumMe and TVSum) demonstrate the superiority of our approach.

## 1 Introduction

The rapid proliferation of videos, driven by ubiquitous recording technologies, social media ecosystems, and streaming platforms, has propelled automated video summarization (VS) to an essential research topic (Alaa et al., 2024; Apostolidis et al., 2021a; Peronikolis & Panagiotakis, 2024; Schiappa et al., 2023). VS involves automatically extracting key parts from source footage, to create a concise overview capturing the semantic essence of the original content. This capability is highly practical, as it enables users to quickly grasp the key points of a video without having to watch the entire footage (e.g., recap lectures (Khetarpaul et al., 2024), filter films (Sharma et al., 2025)).

Recent advances in supervised VS (Son et al., 2024; Narasimhan et al., 2021; Lin et al., 2023; Qiu et al., 2024; He et al., 2023) have yielded compelling results. However, due to the diverse nature of video content and the subjective nature of what constitutes a meaningful summary, the VS datasets (Song et al., 2015; Gygli et al., 2014) are notably limited in size and largely biased in instance distribution, hindering the effectiveness of the SOTA methods for generalization. Therefore, an intuitive, data-driven approach would involve pretraining a foundation video summarizer on the abundance of unlabeled videos and finetuning over supervised data (i.e., user feedback).

To this end, video summarization pretraining methods (Argaw et al., 2024; Narasimhan et al., 2022) generate pseudo ground truth summaries using cross-modality data (e.g., audio or subtitles) and surrogate summarizers (e.g., heuristic rules or LLMs), as shown in Figure 1(a). A video summarizer is then pretrained on these pseudo-summaries and fine-tuned on downstream tasks. However, pretraining on the static pseudo-summaries can be in conflict with the ill-posed nature in video summarization — it may enforce one fixed perspective and overlook the inherent subjectivity and diversity of valid summaries across different viewers and contexts, as exemplified below. In a video of a family picnic, a food vlogger might consider the close-up shots of dishes as key frames for summarization, while a family member might prioritize moments of interaction and laughter. This ill-posed na-

Figure 1: The overall paradigm of existing work (a) and our proposal (b) for video summarization pretraining. We use mutual information to measure how well the summary represents the video.

ture requires pretrained summarizers to capture multifaceted frame importance, so that they can be efficiently fine-tuned to accommodate diverse summarization perspectives in downstream scenarios.

To resolve conflicts between pseudo-summaries and downstream tasks, our idea is in Figure 1 (b): First, pretraining should enable the summarizer to learn how to distinguish more meaningful summaries from unlabeled videos, without perspective differentiation; In this way, finetuning only requires adapting the pretrained multifaceted importance to the specific downstream perspective, facilitating supervised learning. Our pretraining approach, named ViSP, is free of pseudo-summaries, expecting to better align with the ill-posed nature of defining keyframes. Specifically, we first have the summarizer to predict a distribution parameterized by frame score, from which all summaries can be sampled; Then, we use mutual information (MI) to measure which summaries better convey the original video, as MI quantifies their information overlap, needs no annotations (Oord et al., 2018). Subsequently, by optimizing the sampling probability of more meaningful summaries, the summarizer can capture multifaceted importance of frames. Finally, the pretrained model can be fine-tuned to create the state-of-the-art (SOTA) summarizers using the knowledge base behind frame saliency.

We evaluate our proposal through extensive experiments and show that it successfully improves the SOTA summarizers on the SumMe and TVSum benchmarks. Our contributions are threefold:

- We propose a novel pretraining framework that learns versatile frame importance by observing diverse summaries in each unlabeled video.
- We implement the ViSP foundation summarizer, differentiating and exploring more representative summaries through mutual information estimation and learning-based sampling.
- We demonstrate that ViSP is effective in improving SOTA video summarizer performance.

## 2 RELATED WORK

**Video summarization models.** Various model architectures have been proposed to tackle diverse aspects in VS (Alaa et al., 2024; Apostolidis et al., 2021a). They can be broadly categorized as supervised and unsupervised. Many early work focused on non-parametric unsupervised VS (Liu & Kender, 2002; Lu & Grauman, 2013; Potapov et al., 2014) using various heuristics (Kang et al., 2006; Lee et al., 2012; Ngo et al., 2003) and hand-designed features (Ma & Zhang, 2002; Smith & Kanade, 1997; 1995). The introduction of benchmark datasets like TVSum (Song et al., 2015) and SumMe (Gygli et al., 2014) provides frame-level relevance scores from user annotations. This enables the automatic evaluation of video summarization techniques and promotes the burst of supervised learning based methods (Apostolidis et al., 2021b; Rochan et al., 2018; Zhang et al., 2023; Zhu et al., 2020; Arafat & Singh, 2025). These approaches benefit from different neural architectures tailored to video summarization, such as the RNN (Medsker et al., 2001) and LSTM (Hochreiter & Schmidhuber, 1997) modeling variable-range dependencies between frames (Zhang et al., 2016; Wang et al., 2020; Zhao et al., 2018), the CNN (Son et al., 2024; Terbouche et al., 2023) featuring local spatiotemporal relationships, the attention networks (Liang et al., 2022; Fajtl et al., 2019; Ghauri et al., 2021; Fu et al., 2021) such as Transformer (Vaswani et al., 2017) contexualizing all frame (Hsu et al., 2023; Li et al., 2022), the GNN (Zhu et al., 2022; Zhao et al., 2021) better capturing

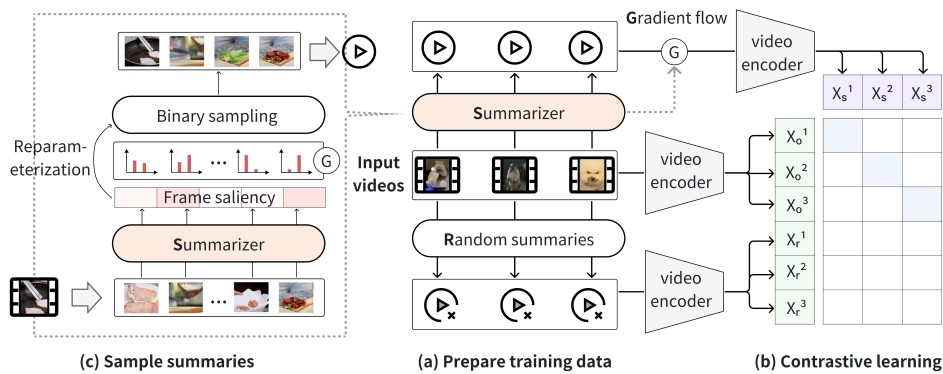

Figure 2: The pretraining workflow of ViSP.

temporal neighbor relationships, etc. A few methods have explored query-focused summarization where users customize the generated summary using a natural language query (Narasimhan et al., 2021; Sharghi et al., 2016; 2017; Kanehira et al., 2018; Akhare & Shinde, 2022). Multimodal summarization (Zhao et al., 2022) has also been considered, where a text input (Plummer et al., 2017; Lin et al., 2023; Qiu et al., 2024) in the form of video captions (Chen et al., 2017) or transcribed speech (He et al., 2023) was incorporated along with the video input to guide video summarization.

In part due to the lack of datasets, many unsupervised variants have been proposed. Model-driven methods benefit from the fast development of deep learning technologies such as GAN (Mahasseni et al., 2017; Apostolidis et al., 2019; 2020), cycle consistent learning objective (Yuan et al., 2019), reinforcement learning (Abbasi et al., 2024; Zhou et al., 2018; Zhang et al., 2019; Zang et al., 2023), and diffusion (Yu et al., 2024). However, these methods struggle to achieve stable and efficient training on unlabeled data across diverse video domains. Data-driven unsupervised approaches benefit from massive unlabeled data and often support supervised variants. As data-driven approaches are most related to our work, we include them in the introduction of pretraining-based related work.

**Pretraining-based video summarization.** We differentiate pretraining-based approaches by leveraging diverse datasets for transfer learning, where models are first trained on auxiliary tasks to learn generalizable representations before being adapted to video summarization. LfVS (Argaw et al., 2024) uses large language models (LLMs) to summarize the speech text and draw corresponding frames as pseudo-summaries for pre-training. TL;DW? (Narasimhan et al., 2022) generates pseudo summaries for pretraining by exploring two heuristic assumptions on instructional videos: (1) key steps repeat across similar videos, and (2) narrators often describe them verbally. SSPVS (Li et al., 2023) pretrains VS model by aligning video-text at multi-granularities while capturing temporal dependency. iPTNet (Jiang & Mu, 2022) makes use of annotated data for moment localization to benefit VS with joint optimization. However, LfVS and TL;DW? rely heavily on pseudo-summaries, often fail to capture the diverse frame importance. The objectives of SSPVS and iPTNet do not directly incentivize VS. In contrast, our method can efficiently learn the versatile frame importance in closed-form optimization and better align with the ill-posed VS goal.

## 3 METHOD

We define the model and use mutual information (MI) to formulate ViSP task in Fig. 1 (b) into one optimization objective (Sec. 3.1). Since MI cannot be directly computed and there are exponentially many summaries that need to be traversed in the objective, we adopt contrastive learning (Sec. 3.2), reparameterization sampling (Sec. 3.3) and extra regularizer (Sec. 3.4) to tackle these challenges.

### 3.1 MODEL AND TRAINING

**Model.** Let $X = \{x_t\}_{t=1}^{T}$ denote a video of $T$ frames, where $x_t$ is a frame feature, such as a raw image or a pre-transformed embedding. Given the original video $X_o$, video summarization aims to select a compact subset $X_s \subseteq X_o$ that optimally represents the content with lowest redundancy.

**For pretraining**, there is no gold summary to supervise $X_s$ in terms of representativeness and redundancy. To this end, ViSP formulates the notion of representativeness using mutual information (MI) and views VS pretraining as an optimization task: Let $\mathcal{I}(X_s; X_o)$ denote the MI between the summary $X_s$ and the original video $X_o$, which measures the amount of information we can obtain for $X_o$ by observing $X_s$ — We search for a small $X_s$ in the possible summaries that maximizes $\mathcal{I}(X_s; X_o)$. To minimize redundancy, we impose the size penalty on $X_s$. Based on the goal, our pretraining objective can be expressed as follows, given the distribution of the video dataset $\mathcal{D}$:

$$\max_{X_s} \quad \mathbb{E}_{X_o \sim \mathcal{D}}[\mathcal{I}(X_s; X_o) - \mathcal{R}(X_s)] \tag{1}$$

where $\mathcal{R}(X_s)$ is regularizer (e.g., size penalty), to avoid trivial solution, such as taking input video as the summary. We also draw a theoretical connection with the information bottleneck (Tishby et al., 2000) in Appendices. However, direct optimization of Eq. (1) is intractable: (1) MI cannot be directly computed; (2) there are $2^T$ candidates for $X_s$ to explore. We will address these challenges with contrastive learning and reparameterization sampling in the following sections.

**For finetuning**, the pretrained summarizer is further finetuned by maximizing the overlap between the predicted summaries $X_s^p$ and ground truth summaries $X_s^g$.

## 3.2 CONTRASTIVE LEARNING FOR MUTUAL INFORMATION ESTIMATION

We can approximate the maximization of mutual information with a contrastive loss, as (Oord et al., 2018) showed that contrastive learning with InfoNCE loss increases a lower bound for MI:

$$\mathcal{I}(X_s; X_o) \geq \log(N) - \mathcal{L}_N \tag{2}$$

where $\mathcal{L}_N$ is the InfoNCE loss, and $N$ indicates the sample size consisting of one positive and $N-1$ negative samples. Note that training samples can be automatically constructed under mini-batch training. As shown in Fig. 2 (a), for each video in the mini-batch, only the summarizer-generated summary is considered positive. To further motivate the generated summary to be informative, a random summary $X_r$ can be used as the hard negative sample. Formally, $\mathcal{L}_N$ is computed as:

$$\mathcal{L}_N = -\sum_{i=1}^{N} \left[ \log \frac{\exp(\text{sim}(X_s^i, X_o^i))}{\sum_{j=1}^{N}[\exp(\text{sim}(X_s^i, X_o^j)) + \exp(\text{sim}(X_s^i, X_r^j))]} \right] \tag{3}$$

here, $X_s^i$ and $X_r^i$ are the generated summary and random summary of the $i$-th video $X^i$ in a mini-batch. The similarity scores are computed via the inner product: $\text{sim}(\cdot, \cdot) = \text{E}(\cdot)^\top \text{E}(\cdot)$, where $\text{E}(\cdot)$ is a video encoder that converts the sequence of frame features $X$ into one vector of dimension $d$.

## 3.3 REPARAMETERIZATION SAMPLING FOR SUMMARY EXPLORATION

**Reformulate summary exploration as sampling.** As the exploration of $2^T$ candidates for $X_s$ is intractable, we consider a relaxation by drawing the summary from a multivariate Bernoulli distribution. To this end, each video frame $x_t \in X_o$ is assigned a binary label $y_t \in \{0, 1\}$: $y_t = 1$ means keep frame $x_t$ in summary $X_s$, discard otherwise. Based on the above assumption, the probability of the summary can be parameterized and factorized:

$$\mathcal{P}_\theta(X_s|X_o) = \prod_{t=1}^{T} \mathcal{P}_\theta(y_t|X_o) \tag{4}$$

where $\theta$ is the parameter of the summarizer. With this relaxation, we can rewrite the objective as:

$$\max_{\theta} \quad \mathbb{E}_{X_o \sim \mathcal{D}} \mathbb{E}_{X_s \sim \mathcal{P}_\theta(X_s|X_o)}[\mathcal{I}(X_s; X_o) - \mathcal{R}(X_s)] \tag{5}$$

**Reparameterization sampling.** We adopt reparameterization method of Concrete-Relaxation (Maddison et al., 2016) to approximate the sampling of discrete binary variables $y_t \sim \mathcal{P}_\theta(y_t|X_o)$:

$$\hat{y}_t = \sigma((\log \epsilon - \log(1 - \epsilon) + \alpha_{\theta,t})/\lambda), \quad \epsilon \sim \text{Uniform}(0, 1) \tag{6}$$

where $\sigma$ is Sigmoid function and $\lambda \in (0, \infty)$ is temperature. There is a zero temperature property in binary concrete relaxation: $\lim_{\lambda \to 0} \mathcal{P}_\theta(\hat{y}_t = 1|X_o) = \frac{exp(\alpha_{\theta,t})}{1+exp(\alpha_{\theta,t})}$. As a result, by choosing

$\alpha_{\theta,t} = \log \frac{\mathcal{P}_\theta(y_t=1|X_o)}{1-\mathcal{P}_\theta(y_t=1|X_o)}$, we have $\lim_{\lambda \to 0} \hat{y}_t = y_t$. This approximation has been proved to have strong rationality (Maddison et al., 2016), such that we use $\hat{y}_t \in (0,1)$ to replace $y_t$ for optimization.

We use the foundation summarizer to generate $\alpha_{\theta,t} \in (-\infty, \infty)$. In cases where the original output isn't compatible, we incorporate an extra output layer for pretraining only. We also consider the advancement in summarizer architecture to ensure the generation of each $\alpha_{\theta,t}$ conditional on all frames in $X_o$, because capturing the holistic context and inter-frame relationships is crucial for accurately identifying key moments. Formally, given summarizer $\Theta$ with parameter $\theta$, we have:

$$\{\alpha_{\theta,t}\}_{t=1}^T = \Theta(X_o) = \Theta(\{x_{o,t}\}_{t=1}^T) \tag{7}$$

We will compare different reparameterization methods in ablation studies (Section 4.3).

**Gradient flow.** To enable gradient flow between $\hat{Y} = \{y_t\}_{t=1}^T$ and sampled frame features, we specify $\hat{X}_s = X_o \odot \hat{Y}$, where $\odot$ denotes gating operation (i.e., frame-wise multiplication). Intuitively, if a particular frame is not important, the corresponding feature takes values close to zero. Finally, the original objective in Eq. (5) is rewritten as follows:

$$\max_\theta \quad \mathbb{E}_{X_o \sim \mathcal{D}} \mathbb{E}_\epsilon [\mathcal{I}(\hat{X}_s; X_o) - \mathcal{R}(\hat{X}_s)] \tag{8}$$

**In some special cases** where the frame features are highly customized — important features may take values close to zero, we can marginalize over the ignored parts as $\mathcal{P}_\theta(X_s) = \sum_{\Delta X} \mathcal{P}_\theta(X_s, \Delta X)$, where $\Delta X$ are ideally sampled from the empirical distribution of the ignored frame features. Inspired by various marginal likelihood estimators (Zintgraf et al., 2017; Ying et al., 2019; Kingma et al., 2013), we can reparameterize $\hat{X}_s$ to approximate $\mathcal{P}_\theta(X_s)$, by sampling a random variable $Z$ from the empirical distribution of frame features:

$$\hat{X}_s = X_o \odot \hat{Y} + Z \odot (\mathbf{I} - \hat{Y}) \tag{9}$$

where Eq. (9) means that we replace masked frame features with frame features taken directly from other videos at the same location. The length of $Z$ can be aligned by up/down sampling the frames.

## 3.4 SATISFYING LENGTH AND BINARY CONSTRAINTS

The framework of ViSP is flexible with various regularization terms to preserve desired properties during summarization. We now discuss the regularization terms as well as their principles. To find a compact summary $X_s$, we apply $l_1$ norm on $\hat{Y}$ by adding $\mathcal{R}_{\text{size}}(\hat{Y}) = ||\hat{Y}||_{l_1}$ as a regularization term. For the binary sampling constraint we consider $\mathcal{R}_{\text{binary}}(\hat{Y}) = (\hat{Y})(1 - \hat{Y})$. To ensure these constraints are satisfied, we optimize them with the Lagrangian function of the overall loss:

$$\mathcal{L}_{total} = \mathcal{L}_N + \beta_{binary} \mathcal{R}_{\text{binary}} + \beta_{size} \mathcal{R}_{\text{size}} \tag{10}$$

where $\beta_{binary}$ and $\beta_{size}$ are Lagrange multipliers corresponding to regularization terms, ensures that constraints are satisfied to what extent by trading off with other losses.

## 4 EXPERIMENTS

We take the SOTA open-source summarizer CSTA as our base, and implement ViSP on top of it. We first detail the experimental settings (Sec. 4.1) and compare the video summarization performance with SOTA summarizers (Sec. 2). Then, we validate our key designs and provide deeper analysis in Sec. 4.3. Finally, we study the manifestations of diversity brought by ViSP (Sec. 4.4).

### 4.1 EXPERIMENTAL SETTINGS

**Metrics and Datasets.** We consider the widely adopted SumMe (Gygli et al., 2014) and TVSum (Song et al., 2015) datasets. SumMe includes 25 videos (1-6 minute) of various themes and camera styles, with summaries created by at least 15 annotators. TVSum contains 50 videos (2-10 minutes) across 10 genres, with 20 annotators assigning shot-level importance scores from 1 to 5. Models aim to match the average human-labeled importance for frames (SumMe) or shots (TVSum). Following recent practices, we evaluate the fine-tuning performance using Kendall's ($\tau$) (Kendall, 1945) and

Table 1: Optional assignments for components and range of values for hyperparameters in our work.

| Variant component | Optional assignment | Hyperparameter | Range |
|---|---|---|---|
| Reparameterization | {Concrete-Relaxation}, Gumbel-Softmax | Tempareture $\lambda$ | (0.05,5) |
| Gradient flow | {Gating}, Marginalization, STE | Penalty $\beta_{size}$ | (0.01,1000) |
| Mutual infomation | {w/o Hard negative}, w/ Hard negative | Penalty $\beta_{binary}$ | (0.01,100) |

Spearman's (Zwillinger & Kokoska, 1999) ($\rho$) coefficients. The $F_1$ score was used previously in video summarization, but is evaluated to be higher if models choose as many short shots as possible and ignore long key shots (Otani et al., 2019; Son et al., 2024; Terbouche et al., 2023). To ensure consistency, we implement five-fold cross-validation on each dataset for train/test splits.

**Implementation.** For fair comparison following He et al. (2023); Zhu et al. (2020); Li et al. (2023); Zhang et al. (2016); Wang et al. (2020), we use frozen pre-trained GoogleNet (Szegedy et al., 2015) to extract frame features as $X \in \mathbb{R}^{T \times d}$ from the corresponding images, where $d = 1024$ is the dimension of frame features. We take CSTA as our base, and implement ViSP on top of it. CSTA takes $X \in \mathbb{R}^{T \times d}$ with one learnable CLS token as input and calculates the importance values $\{\alpha_{\theta,t}\}_{t=1}^{T}$ for $T$ frames. For contrastive learning, we simply use CSTA as video encoder by taking the CLS token in the final hidden state as the video feature. The fine-tuning of the pretrained CSTA is exactly the same as its original training process. We select the best results from five rounds of 5-fold cross-validation to reproduce the CSTA's report in main results. The significance based on all rounds will be analyzed (Section 4.3). Specifically, the pretrained CSTA summarizer is tuned on the mean squared loss by comparing predicted and ground truth scores taken values $(0, 1)$ as follows:

$$\mathcal{L}_{\text{FT}} = \frac{1}{T} \sum_{t=1}^{T} (S_t^p - S_t^g)^2, \quad S_t^p = \sigma(\alpha_{\theta,t}) \tag{11}$$

$\{S_t^g\}_{t=1}^{T}$ are ground truth scores for $T$ frames. During inference, we use the same software as the base model to select summaries based on scores for a fair comparison. For example, CSTA computes the average importance scores of shots into which KTS (Potapov et al., 2014) splits videos, and the summary videos consist of shots with two constraints: $\max \sum S_i^p$ and $\text{Length}_i \leq 15\%$, where $i$ is the index of selected shots. $Length_i$ is the percentage of the length of the $i$-th shot in the original video. Finally, CSTA picks shots with high scores by exploiting the 0/1 knapsack algorithm, and summary videos have a length limit of 15% of the original videos. This ensures that the experimental results will not be contaminated by any carefully designed downstream software used to pick summaries.

**Variants and hyperparameters.** Table 1 specifies the variant components and hyperparameters of ViSP examined in our study. The configurations for the summarizer and fine-tuning remain entirely the same as their original resources for fair, refer to (Son et al., 2024) for details. For each variant consisting of different components, we pre-train on the unlabeled TVSum and SumMe datasets, perform hyperparameter tuning within the range specified in Table 1, and report the best results for each variant. The components in our default variant are enclosed by {}; the code in supplemental provides the random seeds and sampled hyperparameters to reproduce our experiments.

**Baselines.** We compare ViSP with methods of different categories that were discussed in related work. Here are the pretraining-based baselines we mainly compare with: **LfVS** (Argaw et al., 2024) and **TL;DW** based on pseudo-summary alignment, **SSPVS** based on video-text alignment, and **iPTNet** based on moment localization tasks. Both the open-source (**CSTA** (Son et al., 2024)) and closed-source (**LLMVS** (Jiang & Mu, 2022)) SOTA summarizers are considered.

### 4.2 MAIN RESULTS

Table 2 details the experimental results on the SumMe and TVSum benchmarks. We compare ViSP with existing state-of-the-art methods, adhering to their official implementations Except for the methods (Son et al., 2024) combined with ViSP, the baselines are basically classified according to whether transfer tasks are considered, as referred to in related work. In each category, the best-performing baseline is underlined, and the results of our proposal are marked in bold. The finetuning results of **ViSP+CSTA** are based on the same weights pretrained on default configuration. ViSP is successful in providing improvements (up to 3%) over the best open-source model across all metrics. Based on the results, we also make a few comparisons and summarize them as follows.

Table 2: Results on SumMe and TVSum. The official codes for methods* are currently not publicly available for ViSP integration. The baseline results were taken from Son et al. (2024); Argaw et al. (2024); Lee et al. (2025). The human test results were taken from (Otani et al., 2019): due to the subjectivity of the task, the user's rating is not equal to the oracle rating.

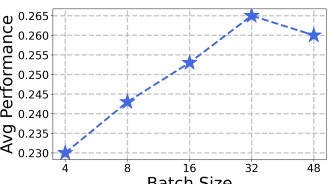

Figure 3: Batch size

| | SumMe | | TVSum | | Avg. |
|---|---|---|---|---|---|
| | $\tau$ | $\rho$ | $\tau$ | $\rho$ | |
| Random | 0.000 | 0.000 | 0.000 | 0.000 | 0.000 |
| Human (Otani et al., 2019) | 0.205 | 0.213 | 0.177 | 0.204 | 0.200 |
| (w/o transfer task) | | | | | |
| SGAN (Mahasseni et al., 2017) | - | - | 0.024 | 0.032 | - |
| DAN* (Liang et al., 2022) | - | - | 0.071 | 0.099 | - |
| CLIP-It* (Narasimhan et al., 2021) | - | - | 0.108 | 0.147 | - |
| STVT (Hsu et al., 2023) | - | - | 0.100 | 0.131 | - |
| PGLSUM (Apostolidis et al., 2021b) | - | - | 0.206 | 0.157 | - |
| AAAM* (Terbouche et al., 2023) | - | - | 0.169 | 0.223 | - |
| MAAM* (Terbouche et al., 2023) | - | - | 0.179 | 0.236 | - |
| VSS-Net* (Zhang et al., 2023) | - | - | 0.190 | 0.249 | - |
| dppLSTM (Zhang et al., 2016) | 0.040 | 0.049 | 0.042 | 0.055 | 0.047 |
| DSNet-AF (Zhu et al., 2020) | 0.037 | 0.046 | 0.113 | 0.138 | 0.084 |
| DSNet-AB (Zhu et al., 2020) | 0.051 | 0.059 | 0.108 | 0.129 | 0.087 |
| DAC* (Fu et al., 2021) | 0.063 | 0.059 | 0.058 | 0.065 | 0.061 |
| DMASum* (Wang et al., 2020) | 0.063 | 0.089 | 0.203 | 0.267 | 0.156 |
| HSA-RNN (Zhao et al., 2018) | 0.064 | 0.066 | 0.082 | 0.088 | 0.075 |
| HMT* (Zhao et al., 2022) | 0.079 | 0.080 | 0.096 | 0.107 | 0.091 |
| RSGN (Zhao et al., 2021) | 0.083 | 0.085 | 0.083 | 0.090 | 0.085 |
| VJMHT* (Li et al., 2022) | 0.106 | 0.108 | 0.097 | 0.105 | 0.104 |
| A2Summ (He et al., 2023) | 0.108 | 0.129 | 0.137 | 0.165 | 0.135 |
| VASNet (Fajtl et al., 2019) | 0.160 | 0.170 | 0.160 | 0.170 | 0.165 |
| MSVA (Ghauri et al., 2021) | 0.200 | 0.230 | 0.190 | 0.210 | 0.208 |
| RR-STG* (Zhu et al., 2022) | 0.211 | 0.234 | 0.162 | 0.212 | 0.205 |
| LLMVS* (Lee et al., 2025) | 0.253 | 0.282 | 0.211 | 0.275 | 0.255 |
| (w/ transfer task) | | | | | |
| iPTNet (Jiang & Mu, 2022) | 0.101 | 0.119 | 0.134 | 0.163 | 0.129 |
| TL:DW (Narasimhan et al., 2022) | 0.111 | 0.128 | 0.143 | 0.167 | 0.137 |
| LfVS* (Argaw et al., 2024) | 0.147 | 0.171 | 0.169 | 0.203 | 0.173 |
| SSPVS (Li et al., 2023) | 0.192 | 0.257 | 0.181 | 0.238 | 0.217 |
| GoogleNet (Szegedy et al., 2015) | 0.176 | 0.197 | 0.129 | 0.163 | 0.166 |
| ViSP+GoogleNet | 0.198 | 0.220 | 0.131 | 0.166 | 0.179 |
| CSTA (Son et al., 2024) | 0.246 | 0.274 | 0.194 | 0.255 | 0.242 |
| **ViSP+CSTA** | **0.273** | **0.305** | **0.201** | **0.263** | **0.260** |

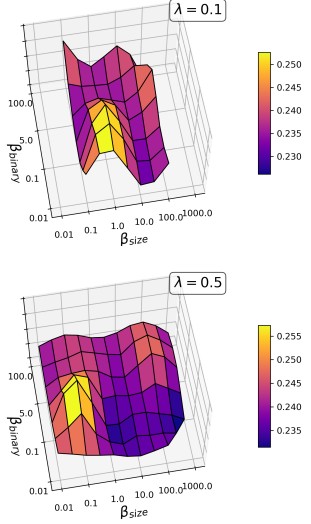

Figure 4: Impact of $\beta_{size}$, $\beta_{binary}$ and $\lambda$ for Concrete-Relaxation.

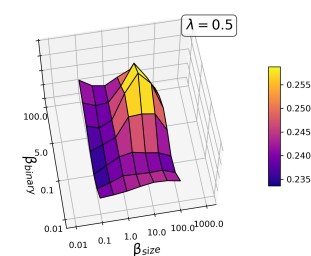

Figure 5: Impact w/o Reparam.

First, baselines using transfer tasks have not benefited from top-notch summarizer architectures, and ViSP outperforms the best summarizer by an average of 4.3%. This might stem from the transfer task being over-coupled with the summarizer structure: iPNet (Jiang & Mu, 2022) relies on modules such as importance propagation and co-teaching to use moment localization data; TL:DW? (Narasimhan et al., 2022) has modules tailored to heuristic assumptions for step perception in instructional videos; SSPVS (Li et al., 2023) requires multimodal data and a text encoder that is aligned with the video encoder at multiple granularities. Only LfVS (Argaw et al., 2024) can adapt to any summarizer by generating pseudo-labels with LLMs, but it has not been open-sourced for further analysis.

Another notable finding is that ViSP can derive advantages from both transfer tasks and the advanced video summarizer. The default variant (i.e., **ViSP+CSTA**) not only achieves an average gain of 1.8% for the best open source summarizer (i.e., CSTA (Son et al., 2024)) but also outperforms the top pretraining-based method (i.e., SSPVS (Li et al., 2023)), registering an improvement of up to 8.1%. Compared with the most competitive closed-source model (i.e., LLMVS (Lee et al., 2025)), our proposal also outperforms 3 out of the 5 reported metrics, including the average performance.

Table 3: Ablation on different proposed components.

| | SumMe | | TVSum | | Avg. |
|---|---|---|---|---|---|
| | $\tau$ | $\rho$ | $\tau$ | $\rho$ | |
| *w/ Concrete and Gating, w/o HardNegative (HN)* | | | | | |
| **ViSP+CSTA** | 0.273 | 0.305 | 0.201 | 0.263 | **0.260** |
| *reparameterization* | | | | | |
| (-) Concrete | **0.276** | **0.308** | 0.192 | 0.252 | 0.257 |
| (+) Gumbel | 0.270 | 0.302 | 0.194 | 0.255 | 0.255 |
| *gradient flow* | | | | | |
| (+) STE | 0.249 | 0.278 | 0.202 | 0.265 | 0.249 |
| (+) Eq. (9) | 0.260 | 0.291 | 0.196 | 0.256 | 0.251 |
| *mutual information* | | | | | |
| (+) HN | 0.248 | 0.277 | **0.204** | **0.267** | 0.249 |
| (+) FSA | 0.241 | 0.269 | 0.192 | 0.252 | 0.239 |

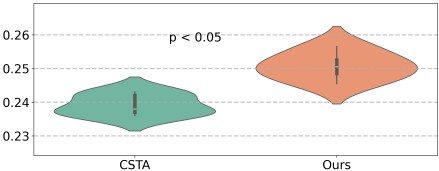

Figure 6: Performance distribution.

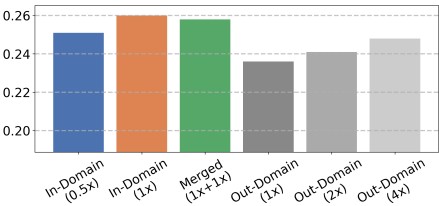

Figure 7: Scales of pretraining data.

### 4.3 ANALYSIS

**Ablation.** We alter each time a component in default ViSP for comparison (Table 3). The reparameterization of *Concrete Relaxation* outperforms that of the *Gumbel* (Jang et al., 2016) on average. *(-) Concrete* (reduces Eq. (6) to Sigmoid) validates the contrastive learning but it biases towards fine-grained SumMe. For CSTA, Eq. (9) may complicate the training, failing to improve *Gating*. Detailed discussion of Eq. (9) is in Appendices. For the impact of binarization, the Straight-Through Estimator (STE) (Bengio et al., 2013) is employed: we apply Bernoulli sampling to $\hat{Y}$, producing binary outputs $\hat{Y}_{\text{binary}}$; then the gradients from $\hat{Y}$ are copied to $\hat{Y}_{\text{binary}}$. The STE improves summarizer on TVSum (shot-level assessment) but degrades on SumMe (frame-level evaluation). This discrepancy suggests that full binarization is more effective at capturing coarse-grained saliency patterns. The same phenomenon can also be observed when using random summaries as hard negatives (HN).

**Impact of 4 hyperparameters in ViSP.** The batch size $N$ is related to MI estimation Eq. (3), while size penalty $\beta_{\text{size}}$, binarization penalty $\beta_{\text{binary}}$, and temperature $\lambda$ are associated with reparameterized sampling. As shown in Fig. 3, when $N$ is set too small (e.g., $N = 4$), the MI lower bound in Eq. (2) becomes overly relaxed, impairing CSTA's performance. To benefit CSTA from pretraining, $N \geq 8$ is recommended. Fig. 4 shows that as the $\lambda$ decreases (from 0.5 to 0.1), the optimal $\beta_{\text{binary}}$ decreases while $\beta_{\text{size}}$ increases. We attribute this to the fact that reducing the $\lambda$ also promotes binarization, but requires a stronger size penalty to prevent the model from trivially generating long summaries. To validate this, we analyze the impact of hyperparameters without reparameterization in Fig. 5, where Eq. (6) reduces to Sigmoid. As the Sigmoid is continuous and differentiable, $\lambda$ primarily affects the gradient magnitude rather than approximating discrete sampling (Maddison et al., 2016). As a result, the sigmoid relies on stronger regularization penalties to bring improvement.

**Statistical significance analysis.** Figure 6 shows the performance distribution of CSTA and ViSP, where Welch's *t*-test was performed to compute the *p*-values. The results demonstrate that ViSP's improvement over CSTA is statistically significant, *p*-values $< 0.05$. In addition, since most baselines only report the average results of five-fold cross-validation, we can only compare the significance of performance differences with the SOTA open-source method (i.e., CSTA).

**Scaling pretrainset and unseen setting.** In Fig. 7, we analyze the distribution shifts or scales of pretraining data. The vertical axis represents ViSP's average performance on SumMe and TVSum. We first use the full unlabeled datasets of SumMe and TVSum as *In-Domain* data, establishing the baseline scale (1×). Through random sampling, we obtain in-domain data at $0.5\times$ scale, which provides a setting where the test videos are not seen during pre-training. Subsequently, we randomly select out-of-domain videos from YouTube (De Avila et al., 2011), OVP (De Avila et al., 2011), and ActivityNet (Fabian Caba Heilbron & Niebles, 2015), ensuring they are distinct from SumMe and TVSum, to construct *Out-Domain* datasets. Figure 7 reveals that *In-Domain* pretraining scaling brings maximal downstream gains. Though Out-Domain pretraining underperforms at equal scales, it can be improved through scaling. This indicates that pretraining indeed learns transferable representations, where the domain distribution of the dataset also matters.

Table 4: Entropy of per-frame selection probability (EPFSP) & summarizer performance versus $\lambda$.

| Metric | $\lambda = 0.005$ | $\lambda = 0.01$ | $\lambda = 0.05$ | $\lambda = 0.5$ | $\lambda = 1.0$ | $\lambda = 5.0$ |
|---|---|---|---|---|---|---|
| EPFSP | 0.004 | 0.008 | 0.042 | 0.339 | 0.501 | 0.677 |
| Avg. $\tau/\rho$ | 0.239 | 0.244 | 0.256 | 0.260 | 0.249 | 0.234 |

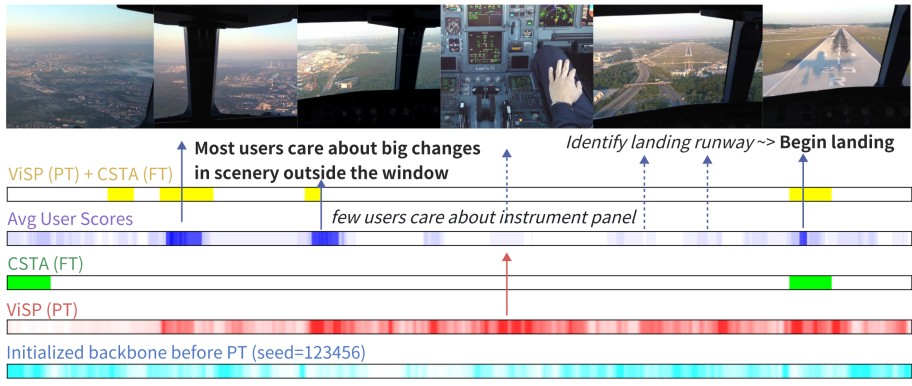

Figure 8: The images and normalized frame scores from a piece of video titled "Cockpit Landing". The color for all highest frame score is enhanced. The pretrain/finetune process is denoted as PT/FT.

## 4.4 DIVERSITY STUDIES

**Is the mechanism of ViSP sampling diverse summaries useful?** We replaced the diverse summaries in ViSP with high-quality fixed pseudo-summaries for pretraining, and ablation results are in **(+) FSA** of Table 3. Since the prior pseudo summaries is not publicly available, we use ground truth labels to construct high-quality pseudo-summaries, forming a strong FSA-based baseline. We use label to select the top 15% of the TVSum video as FSA based on Eq. (13); then, we train CSTA on FSA with binary classification loss, and report the finetuning results on SumMe. Similarly, we constructed FSA from SumMe for pretraining and fine-tuned CSTA on TVSum. Note that, unlike FSA, ViSP does not access any label. Though fixed pseudo-summaries are of high quality (drawn from human labels), pretraining on them may hinder CSTA performance, as the model could overfit to fixed-length and fixed-perspective summaries, compromising generalization.

**Changes of frame saliency after pretraining.** Qualitative results in Fig. 8 show that the summary generated by ViSP+CSTA covers more ground truth frames of high human bids. By comparing the two saliency bars at the bottom, we observed that without finetuning, the frame saliency obtained solely through ViSP pre-training is clearly concentrated in the frames with user bids. Even frames preferred by niche users are also taken into account by ViSP. More cases can be found in Appendices.

**Relationship between $\lambda$ and pretraining diversity.** We reported the entropy of per-frame selection probability after pretraining (diversity) and summarization performance after finetuning versus $\lambda$ in Table 4. Diversity increases monotonically with $\lambda$, but there is no monotonicity between the diversity and average $\tau/\rho$ correlations and there is a tradeoff. Excessively high diversity means that redundant or noisy frames are difficult to distinguish. Excessively low diversity causes the pretrained distribution to fail to cover potential perspectives, similar to another type of fixed pseudo-summary.

## 5 CONCLUSION

We considered video summarization pretraining and introduced ViSP, a pretraining framework that automatically learns the versatile frame importance from unlabeled raw videos. Unlike dominant pretraining methods that rely on static pseudo-summaries, ViSP can efficiently dynamically explore diverse summaries and measure their utilities. This addresses the issue that static pseudo-summaries poorly align with the ill-posed nature of defining keyframes. Extensive experiments demonstrate our superiority. Extra results, analysis and codes can be found in the **Appendices/Supplemental**.

STATEMENT

**Ethics statement.** The algorithm we propose does not raise new Ethics concerns, but may inherit the internal biases of the training data.

**Reproducibility statement.** We provide in the Experimental Section and Appendices a clear setup for reproducibility. We also upload the code for pretraining and finetuning, as well as the checkpoint of the main experiment as supplemental materials. This ensures reproducibility.

**Use of LLMs in writing.** We only use LLMs to polish writing, e.g., grammar/spelling checking. We also double-check the polished texts to try our best to optimize the readers' experience.

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

# A APPENDICES

## A.1 DATASET DETAILS

We use two standard video summarization datasets: **SumMe** contains videos with diverse contents (such as holidays, events, sports, etc.) and various types of shooting angles (such as egocentric, moving and static). These videos are either raw or edited public videos, with a duration of 1 to 6 minutes. At least 15 people have created ground truth summary videos for all the data, and these models have predicted the average number of selections made by people for each frame. **TVSum** contains 50 videos from 10 genres (such as documentaries, news, and vlogs). These videos have a duration of 2 to 10 minutes, and 20 people have annotated the ground truth for each video. The ground truth is the importance score at the shot level, ranging from 1 to 5, and the model attempts to estimate the average score at the shot level. The relevant information is presented in Table 5.

We also include additional data to analyze the scales of pretraining data: YouTube (De Avila et al., 2011), OVP (De Avila et al., 2011), and ActivityNet (Fabian Caba Heilbron & Niebles, 2015). YouTube has 39 videos. These videos are distributed among several genres (cartoons, news, sports, commercials, tv-shows and home videos) and their duration varies from 1 to 10 min. The OVP contains 50 videos from Open Video Project. All videos are in MPEG-1 format (30 frames per second, 352×240 pixels), in color and with sound. These videos are categorized into multiple genres (documentary, educational, ephemeral, historical, lecture), with durations ranging from 1 to 4 minutes, and the total video duration is approximately 75 minutes. ActivityNet provides samples of 203 activity categories across 7 major categories (such as Household, Caring and Helping, Personal Care, etc.), with an average of 137 untrimmed videos per activity category, 1.41 activity instances per video, and a total video duration of 849 hours. All datasets are publicly available under open license agreements. TVSum follows the Creative Commons CC-BY (v3.0) License. SumMe adheres to the research-only terms. YouTube and OVP are licensed under the MIT License. ActivityNet is also governed by the MIT License.

Table 5: Dataset overviews.

|  | **SumMe** | **TVSum** | **YouTube** | **OVP** | **ActivityNet** |
|---|---|---|---|---|---|
| Source | Youtube | Youtube | Youtube | Open Video Project | Youtube |
| Number of Data | 25 | 50 | 39 | 50 | 27801 |
| Total Video Duration(Hours) | 1.1 | 3.5 |  | 1.3 | 849 |
| Mean Video Duration(mins) | 2.7 | 4.2 |  | 1.5 | 1.83 |
| Max Video Duration(mins) | 6.5 | 10.8 | 10.0 | 4.0 |  |
| Min Video Duration(mins) | 0.7 | 1.4 | 1.0 | 1.0 |  |
| Number of Classes | 25 | 10 | 6 | 5 | 7 |

## A.2 BASE MODEL DETAILS

We base our pretraining framework on **CSTA** (Son et al., 2024) summarizer, which is released under the MIT License. CSTA is a CNN-based spatio-temporal attention method. This approach stacks the features of each frame in a single video to form image-like frame representations, and applies a 2D convolutional neural network (2D CNN) to these frame features. The method relies on CNN to understand the inter-frame and intra-frame relationships, and to mine the key attributes in videos by leveraging its ability to learn absolute positions within images. Unlike previous works that sacrifice efficiency by designing additional modules to focus on spatial importance, CSTA uses CNN as a sliding window, requiring minimal computational overhead. CSTA uses a pre-trained and frozen GoogleNet (Szegedy et al., 2015) to obtain the image representation of each frame. The features will be appended with a CLS token shaped as 3×1,024. Meanwhile, GoogleNet is employed as a trainable CNN to match the dimension of all features to 1,024. The encoded features go through the attention module and the mixing module before being fed into the classifier. Based on the mixed features output by the mixing module, the classifier generates importance scores. All CNN models are pre-trained on ImageNet. The initial weights of the linear layers in the classifier are initialized via Xavier initialization, while the key and value embeddings are initialized randomly. Both the output channels of the linear layers and the embedding dimensions of keys and values are 1,024.

## A.3 IMPLEMENTATION DETAILS

Following (He et al., 2023; Zhu et al., 2020; Li et al., 2023; Zhang et al., 2016; Wang et al., 2020) for a fair comparison, we use frozen pre-trained GoogleNet (Szegedy et al., 2015) to extract frame features as $X \in \mathbb{R}^{T \times d}$ from the corresponding images, where $d = 1024$ is the dimension of frame features. We take CSTA as our base, and implement ViSP on top of it. CSTA takes $X \in \mathbb{R}^{T \times d}$ with one learnable CLS token as input and calculates the importance values $\{\alpha_{\theta,t}\}_{t=1}^{T}$ for $T$ frames. For contrastive learning, we simply use CSTA as video encoder by taking the CLS token in the final hidden state as the video feature. Notably, there are two CSTA models currently: one used as summarizer model, one used as video encoder. After pretraining, the CSTA model used as video encoder is discarded. The fine-tuning of the pretrained CSTA is exactly the same as its original training process. We select the best results from five rounds of 5-fold cross-validation to reproduce the CSTA's report in main results. The significance based on all rounds will be analyzed (Section 4.3). Specifically, the pretrained CSTA summarizer is tuned on the mean squared loss by comparing predicted and ground truth scores taken values $(0, 1)$ as follows:

$$\mathcal{L}_{\text{FT}} = \frac{1}{T} \sum_{t=1}^{T} (S_t^p - S_t^g)^2, \quad S_t^p = \sigma(\alpha_{\theta,t}) \tag{12}$$

$\{S_t^g\}_{t=1}^{T}$ are ground truth scores for $T$ frames. For inference, CSTA creates summary videos based on shots that KTS (Potapov et al., 2014) derives. It computes the average importance scores of shots into which KTS splits videos. The summary videos consist of shots with two constraints:

$$\max \sum S_i^p, \qquad \text{Length}_i \leq 15\% \tag{13}$$

where $i$ is the index of selected shots. $Length_i$ is the percentage of the length of the ith shot in the original videos. CSTA picks shots with high scores by exploiting the 0/1 knapsack algorithm, and summary videos have a length limit of 15% of the original videos.

Pretraining for 200 epochs with a batch size of 32 takes approximately 30 minutes on 75 in-domain videos (will be analyzed later in Section 4.3.4) and requires around 30GB of GPU memory. A five-fold cross-validation fine-tuning experiment with a batch size of 1 takes about 2 hours and uses roughly 3GB of GPU memory. Note that batch size influences the estimation of mutual information Eq. (2), but does not affect the finetuning objective Eq. (11).

## A.4 EXTRA ANALYSIS

**Summarization of very long video.** QFVS (Sharghi et al., 2017) provides hour-long videos that can be used to evaluate the proposed method with longer videos. Yet, during processing, we found that it is too long. Running KTS for QFVS evaluation takes 4,000 hours (OOT) for each video, and the number of frames in a video also exceeds the processing capacity of the summarizer. Therefore, we have made some modifications to the standard experimental procedures provided by the benchmark in order to obtain referable results. We extract 5,000 frames from each video to form a new original video, on which we perform video summarization pretraining and finetuning. Finally, we combine the importance scores generated separately for each segment and directly calculate the correlation coefficients without using KTS (OOT) for shot selection. The results of the leave-one-out experiment show that ViSP brings a 2-fold performance improvement to CSTA as shown in Table 6. ViSP aims to benefit the base summarizer by using unlabeled video data. Despite adopting the simplest scheme, the experimental results show that ViSP can also improve the CSTA on long videos.

Table 6: Generalizability of CSTA+ViSP on long videos (QFVS).

| Model | CSTA | CSTA+ViSP pretrained on TVSum+SumMe | CSTA+ViSP pretrained on TVSum+SumMe+QFVS |
|---|---|---|---|
| QFVS-$\tau$ | $0.030 \pm 0.002$ | $0.044 \pm 0.011$ | $0.084 \pm 0.014$ |
| QFVS-$\rho$ | $0.036 \pm 0.002$ | $0.050 \pm 0.012$ | $0.075 \pm 0.014$ |

**Human results.** The phenomenon where model results surpass individual human scores has been discussed in several works (Son et al., 2024; Argaw et al., 2024; Otani et al., 2019). As noted in

(Otani et al., 2019), video summarization is inherently ill-posed: while humans often provide subjective, diverse summaries, current evaluation metrics favor alignment with the statistical average, which benefits models trained to mimic this average.

**Theoretical connection with the information bottleneck (IB).** IB (Tishby et al., 2000) seeks a compressed representation $Z$ of an input $X$ by maximizing $I(Z;Y) - \beta \cdot I(X;Z)$, balancing information about a target $Y$ (sufficiency) against compression (minimality). In ViSP's self-reconstruction setting, the original video $X_o$ acts as both the input $X$ and the target $Y$ (i.e., $Y \equiv X_o$), while the summary $X_s$ is the compressed representation $Z$. Therefore, maximizing the relevance term $I(X_s; X_o)$ directly corresponds to maximizing $I(Z;Y)$, ensuring the summary is *sufficient*. The regularization term $R(X_s)$, which penalizes summary length, serves as a proxy for minimizing the compression term $I(X;Z)$, thus enforcing *minimality* and creating the bottleneck. This reframes the task as finding a minimal sufficient self-representation of the video.

**Transferability across model architectures.** To further investigate the transferability of the proposed pretraining approach, we conducted experiments. For Transformer, we use the standard transformer provided by pytorch.nn, with a depth of 6, 8 num heads, a hidden layer dimension of 512; for GNN, we generate edges for adjacent frames and use GCNConv provided by torch-geometric to construct the GNN, with a depth of 5, a hidden layer dimension of 2048; dropout is set to 0.1, and finally there is an additional linear layer to project each hidden feature to 1 to generate the frame score. Table 7 indicates that our pretraining approach proves effective for the presented architectures, with $p$-values $< 0.05$ for all improvements.

Table 7: Results across different model architectures.

| Model | SumMe-$\tau$ | SumMe-$\rho$ | TVSum-$\tau$ | TVSum-$\rho$ | Avg. |
|---|---|---|---|---|---|
| CSTA | 0.246 | 0.274 | 0.194 | 0.255 | 0.242 |
| CSTA+ViSP | 0.273 | 0.305 | 0.201 | 0.263 | 0.260 |
| GoogleNet | 0.176 | 0.197 | 0.129 | 0.163 | 0.166 |
| GoogleNet+ViSP | 0.198 | 0.220 | 0.131 | 0.166 | 0.179 |
| Transformer | 0.184 | 0.205 | 0.087 | 0.106 | 0.146 |
| Transformer+ViSP | 0.209 | 0.234 | 0.148 | 0.195 | 0.196 |
| GNN | 0.175 | 0.195 | 0.059 | 0.067 | 0.124 |
| GNN+ViSP | 0.187 | 0.208 | 0.128 | 0.169 | 0.173 |

**IoU Test of sampled summaries.** For completeness, we also compute the pairwise IoU over five sampled summaries per video and report the averaged results in table 8 as a reference for some summarization size. We found that the pairwise IoU computed from hard-sampled summaries depends on an ill-defined summarization size, making it difficult to analyze reliably.

Table 8: Summarization size versus pairwise IoU between 5 hard-sampled summaries.

| Size | $\lambda = 0.005$ | $\lambda = 0.01$ | $\lambda = 0.05$ | $\lambda = 0.5$ | $\lambda = 1.0$ | $\lambda = 5.0$ |
|---|---|---|---|---|---|---|
| 10% | $0.1107 \pm 0.0128$ | $0.1085 \pm 0.0132$ | $0.1079 \pm 0.0143$ | $0.0818 \pm 0.0122$ | $0.0692 \pm 0.0115$ | $0.0522 \pm 0.0099$ |
| 30% | $0.4298 \pm 0.0363$ | $0.4244 \pm 0.0330$ | $0.4137 \pm 0.0420$ | $0.2849 \pm 0.0150$ | $0.2335 \pm 0.0150$ | $0.1802 \pm 0.0100$ |
| 60% | $0.8233 \pm 0.0559$ | $0.8746 \pm 0.0559$ | $0.9687 \pm 0.0118$ | $0.6691 \pm 0.0237$ | $0.5406 \pm 0.0164$ | $0.4343 \pm 0.0081$ |
| 90% | $0.8376 \pm 0.0055$ | $0.8406 \pm 0.0057$ | $0.8767 \pm 0.0094$ | $0.9401 \pm 0.0089$ | $0.8921 \pm 0.0099$ | $0.8191 \pm 0.0039$ |

**Performance gain without KTS+knapsack.** We conducting an ablation study on KTS+knapsack for further strengthen the conclusions. In particular, KTS and knapsack selection are used to generate shot-level scores, allowing the summarizer to be evaluated on the shot-level labels provided by TVSum. In fact, skipping KTS and knapsack is equivalent to evaluating frame-level relevance. The frame-level correlation coefficient without KTS+knapsack is shown in table 9.

Table 9: Ablation results on KTS+knapsack.

| w/o KTS+knapsack | SumMe-$\tau$ | SumMe-$\rho$ | TVSum-$\tau$ | TVSum-$\rho$ | Avg. |
|---|---|---|---|---|---|
| CSTA | $0.133 \pm 0.013$ | $0.181 \pm 0.017$ | $0.334 \pm 0.006$ | $0.479 \pm 0.006$ | $0.282 \pm 0.008$ |
| CSTA+ViSP | $0.163 \pm 0.002$ | $0.219 \pm 0.002$ | $0.342 \pm 0.006$ | $0.487 \pm 0.009$ | $0.303 \pm 0.004$ |

**Extra Results on OOD Scaling.** For the experiment in Fig. 7, after we initially collected and constructed out-of-domain (OOD) data, we filtered in-domain videos based on video duration and titles, and randomly split them. We designed experiments as follows. To further strengthen the conclusion, we supplement the domain distance versus performance gains. We calculate the L2 distance between the average frame feature of each out-of-domain video and the average frame feature of the nearest neighbor in-domain video, then sort all out-of-domain videos, and divide them equally into three sets for pre-training. The nearest in-domain distance is calculated based on the two closest data points between the OOD split and the in-domain split. There is no need to specifically remove near-duplicate videos because we can quantify distribution shifts based on the nearest in-domain distance. In-domain data is the unlabeled training set in TVSum, SumMe benchmarks.

Table 10: Results on Out-of-Domain Scaling controlled by nearest in-domain distance.

| Metric | OOD split-1 | OOD split-2 | OOD split-3 |
|---|---|---|---|
| nearest in-domain distance | 0.314 | 0.537 | 0.608 |
| Avg. performance | 0.252 | 0.245 | 0.23 |

**Input Feature of Video Encoder Should be the Same as that of the Summarizer.** ViSP is a relaxed implementation of the objective function (Eq. (1)), grounded in an information-bottleneck (line 172 and Appendices 4.3). This indicates that $X_s$ is searched in the feature space defined based on $X_o$. Changing the feature space where the video encoder is located, the perception of information and the search for summaries occur in different spaces, and the pre-trained reward and fine-tuned reward are also based on different feature spaces respectively, resulting in deviation. Table 11 shows that the mutual information estimation performs better when the features used by the video encoder and the summarizer are consistent. Intuitively, even if the video encoder uses richer information to improve the mutual information estimation, the summarizer cannot utilize this additional information for summarization, resulting in a gap.

Table 11: Ablation results on video encoder

| Video Encoder | SumMe-$\tau$ | SumMe-$\rho$ | TVSum-$\tau$ | TVSum-$\rho$ | Avg. |
|---|---|---|---|---|---|
| CSTA | 0.273 | 0.305 | 0.201 | 0.263 | 0.260 |
| VideoPrism | 0.246 | 0.275 | 0.197 | 0.259 | 0.244 |

**F1 Metrics.** We supplement the F1 score for reference in Table 12. ViSP has also achieved improvement in the F1 score, which demonstrates its effectiveness. Below we provide a complete academic background. Before the work (Otani et al., 2019) analyzed the limitations of F1, video summarizers were mainly optimized by calculating cross-entropy loss based on the binary chosen labels of frames. After the work (Otani et al., 2019), using MSE loss to fit the evaluation scores, and finally evaluating the correlation coefficients after KTS + knapsack processing are commonly used.

Table 12: Traditional F1 Metric for reference.

| Method | SumMe-F1 | TVSum-F1 |
|---|---|---|
| CSTA | $0.537 \pm 0.005$ | $0.557 \pm 0.002$ |
| CSTA+ViSP | $0.570 \pm 0.010$ | $0.574 \pm 0.004$ |

**The Relationship Between Temperature, Pretraining Diversity, and FineTuning Performance.** We introduce EPFSP (Entropy of Per-Frame Selection Probability) to verify the diversity of the sampling process for video summarization as shown in Table 4. The pretraining configuration is the same as the main experiment in the paper (Table 2). We found that as $\lambda$ increases, the entropy of the per-frame selection probabilities (i.e., diversity) increases. However, there is no monotonic relationship between the diversity metric and Kendall/Spearman correlations and there is a tradeoff. Excessively high diversity means that redundant or noisy frames are difficult to distinguish. Excessively low diversity causes the pretrained distribution to fail to cover potential perspectives, similar to another type of fixed pseudo-summary.These conclusions are consistent with Fig. 1, lines 72-82

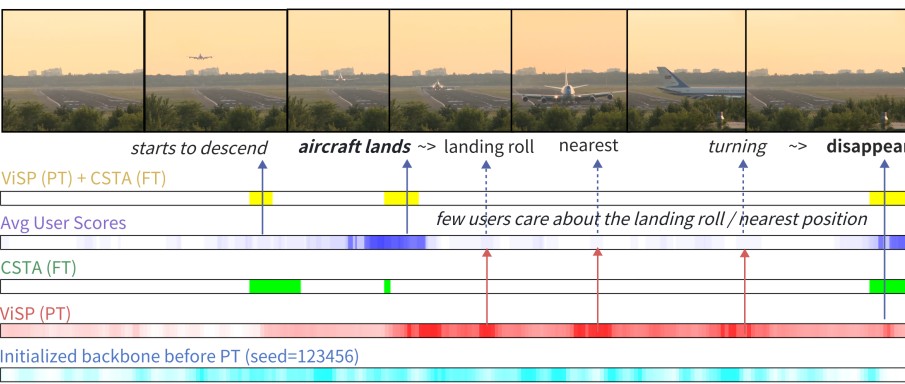

Figure 9: The images and normalized frame scores from a video titled "Air Force One". The color for all highest frame score is enhanced. The pretrain/finetune process is denoted as PT/FT.

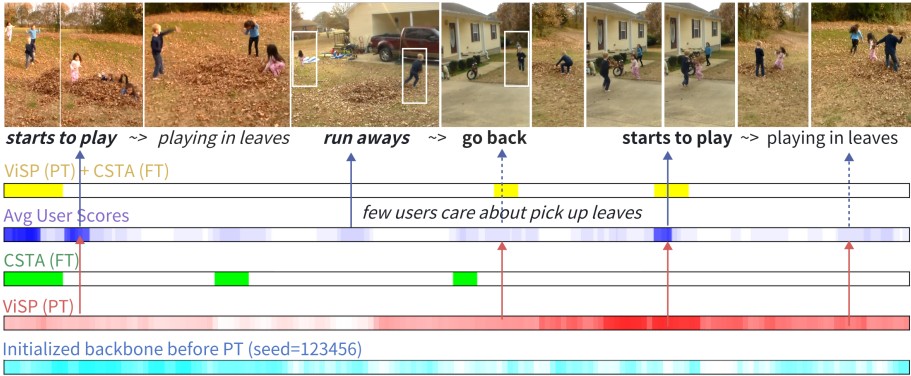

Figure 10: The images and normalized frame scores from a video titled "Kids playing in leaves". The color for all highest frame score is enhanced. The pretrain/finetune process is denoted as PT/FT.

and Section 4.4 of the paper. Pretraining aims to remove more redundancy while retaining useful information to cover as many potential perspectives as possible; there is a tradeoff. In this way, fine-tuning only requires selecting perspectives, making it easier to learn. We also tested the coverage between summaries randomly sampled from the pre-trained distribution and user summaries. The coverage between the sampled summary and the best user summary reaches $80\% \sim 92\%$ when pre-trained summaries retain on average $62\%$ of the frames from the original videos (vs. $15\%$ retained after fine-tuning). These statistics suggest that pre-training enables the model to retain core content from multiple perspectives altogether, requiring finetuning to distinguish one preferred perspective.

**More case studies.** We provide two additional cases to strengthen the conclusions of Section 4.4. Generally speaking, we usually take the transitions of action states as key frames to be included in the summary, because the duration of action states is usually highly repetitive in semantics and easy to predict. In Figure 10 and Figure 9, we have specifically bolded the text indicators of the transitions of action states. We found that after ViSP pre-training, the fine-tuned CSTA is easier to discover these key frames. However, there are also failed cases. For example, in Figure 10, neither CSTA nor ViSP+CSTA selected the corresponding shot of "run aways". This indicates that the video summarization still has room for improvement.

**Discussion on contribution.** The similarity score in ViSP is not an independent module, but is learned through contrastive learning, as shown in Eq. (3). Notably, the VS pre-training is an un-supervised process without ground truth labels, making it unable to provide a correct direction for the direct optimization of similarity measures. Additionally, our work does not aim to advance deep

learning theory per se. Our contribution lies in introducing a novel pre-training paradigm for VS and demonstrating its efficacy through practical modeling. While we leverage concepts such as mutual information and differentiable sampling, these serve to theoretically support the proposed approach rather than constitute standalone theoretical contributions. So these theoretical components are not a weakness. To our knowledge, existing video-summarization pre-training frameworks focus on constructing pseudo-summaries and largely overlook diversity. We therefore present the first adaptation of a pre-training framework that explicitly models this ill-posed, diversity-sensitive task, which has already sparked new discussions, such as explaining or demonstrating the intuitive changes in diversity for different needs in pre-training blackbox. In future work, we plan to explore the interpretability of the pre-training process to further address these challenges. We believe these ongoing discussions enrich the field and do not undermine the validity of our current contributions.

## A.5 DEEPER ANALYSIS OF Z AND EQ. (9)

For the convenience of discussion, we consider the mask formulation of each frame.

- Multiplicative masking (i.e., gating): $\hat{x} = x \cdot y$
- Marginalization-based masking (Eq. (9)): $\hat{x} = x \cdot y + (1 - y) \cdot z$

For each frame $x \in X_o$, we have corresponding frame-level mask value $y \in \hat{Y}$ and auxiliary variable $z \in Z$. Assume frame features corresponding to different semantics follow Gaussian distributions. For two distinct frames $x_1 \sim q_1(x_1)$, where $q_1(x_1) = \mathcal{N}(\mu_1, \sigma_1^2)$, and $x_2 \sim q_2(x_2)$, where $q_2(x_2) = \mathcal{N}(\mu_2, \sigma_2^2)$, gating with a small $y$ yields:

$$\hat{x}_1 \sim \mathcal{N}(y\mu_1, y^2\sigma_1^2) = q_1(y \cdot x_1) \quad \text{and} \quad \hat{x}_2 \sim \mathcal{N}(y\mu_2, y^2\sigma_2^2) = q_2(y \cdot x_2).$$

These distributions still preserve the same semantic distinctions when $y \neq 0$ because:

$$D_{\mathrm{KL}}(q_1(x_1)\|q_2(x_2)) = D_{\mathrm{KL}}(q_1(y \cdot x_1)\|q_2(y \cdot x_2))$$

for Gaussian distributions. Eq. (9) introduces an auxiliary variable $z \sim p_z$ specifically to eliminate this residual discriminability by reducing the distance between distributions $p(\hat{x}_1)$ and $p(\hat{x}_2)$ when $y \to 0$.

**Theorem 1** *Let $x_1 \sim q_1$, $x_2 \sim q_2$ and fix $0 < y < 1$. Let $z$ (and $z'$) have law $p_z$. Define:*

$$v = yx_1 + (1 - y)z, \qquad w = yx_2 + (1 - y)z'$$

*Let $p_{yx_i}$ denote the law of $yx_i$, and $p_v, p_w$ denote the laws of $v, w$. Then:*

$$D_{\mathrm{KL}}(p_v\|p_w) \leq D_{\mathrm{KL}}(p_{yx_1}\|p_{yx_2})$$

**Proof 1** *We view the construction of $v, w$ as two successive Markov kernels.*

*(1) **Deterministic scaling** $S : x \mapsto yx$. For $i = 1, 2$ write $q_i$ for the law of $x_i$. Then the law after scaling is $p_{yx_i} = q_i S$.*

*(2) **Additive-noise** is kernel $K$: Given $s$ output $s + (1 - y)z$. This kernel $K(\cdot \mid s)$ is the same for both inputs. Applying $K$ to $p_{yx_i}$ yields $p_v = (q_1 S)K$ and $p_w = (q_2 S)K$.*

*(3) Use the **data-processing property** of $D_{\mathrm{KL}}$. For any two input distributions $(P, Q)$ and any Markov kernel $K$, $D_{\mathrm{KL}}(PK\|QK) \leq D_{\mathrm{KL}}(P\|Q)$. Apply this with $P = q_1 S$, $Q = q_2 S$ and kernel $K$, we have:*

$$D_{\mathrm{KL}}\big((q_1 S)K \,\big\|\, (q_2 S)K\big) \leq D_{\mathrm{KL}}(q_1 S\|q_2 S)$$

*If the noise kernel $K$ is nontrivial and $p_{yx_1} \neq p_{yx_2}$, the inequality is typically strict: adding noise via the same channel reduces distinguishability.*

When $y \to 1$, $\hat{x} \to x$; when $y \to 0$, $\hat{x} \to z \sim p_z$, removing semantic cues. In Section 4.3 we find that naively using real frame features as $z$ reduces pretraining benefits. We attribute this to:

(i) the backbone features are not parameterized by challenge distribution thus gating-based mask can easily handle;

(ii) directly using features from other videos as $z$ keeps them in the same distribution as unmasked frames, which may confuse the encoder.

To fairly verify the utility of Eq. (9), we adopt a controlled setting with $h \sim \mathcal{N}(x, 1)$ as frame features and $z \sim \mathcal{N}(0, 1)$. Experiments show that Eq. (9) consistently outperforms simple multiplicative masking under this controlled setting. The results are reported in Table 13.

Table 13: Results on controlled setting, where we take $h \sim \mathcal{N}(x, 1)$ as frame features.

| Method | SumMe-$\tau$ | SumMe-$\rho$ | TVSum-$\tau$ | TVSum-$\rho$ | Avg. |
|---|---|---|---|---|---|
| CSTA | $0.203 \pm 0.004$ | $0.227 \pm 0.004$ | $0.087 \pm 0.003$ | $0.114 \pm 0.003$ | $0.158 \pm 0.003$ |
| CSTA+ViSP | $0.231 \pm 0.003$ | $0.258 \pm 0.004$ | $0.074 \pm 0.010$ | $0.098 \pm 0.014$ | $0.165 \pm 0.007$ |
| w/ Eq. (9) | $0.233 \pm 0.011$ | $0.259 \pm 0.013$ | $0.086 \pm 0.002$ | $0.114 \pm 0.002$ | $0.173 \pm 0.005$ |

We can see similar parameterization of image features using normal distribution in VAE (Kingma & Welling, 2013). However, currently advanced summarizers are not based on such features, so we adopted the simplest adaptation as a reference to support the extra analysis.

## A.6 LIMITATIONS

Our research focuses on video summarization pretraining but shares existing frameworks' structural limitations. First, we primarily use open-source CSTA for experiments. While effective, the analysis of performance upper bound is constrained by its architecture and parameter scales; this can be improved when more SOTA summarizers are publicly available. Second, the generated summary might inherit biases present in the original video, which could affect fairness when applied to diverse populations or sensitive contexts. Addressing this requires strategies for fair, interpretable outcomes from complex models, presenting a promising research area. Finally, the method's performance may be sensitive to the choice of pretraining datasets and tasks, as our experiments show that the in-domain transferability of the learned representations is better.

## A.7 BROADER IMPACTS

While VS pretraining can enhance efficiency in content analysis and accessibility, there are several potential negative societal impacts. First, since unlabeled videos haven't been effectively reviewed, the summarizer may learn about potential harmful content in the pre-trained data. Second, our approach could be misused to selectively omit or distort critical information in summaries, propagating bias or disinformation. Additionally, automated summarization deployed in surveillance contexts could raise privacy concerns if sensitive details are inadvertently retained or misrepresented. To mitigate these risks, further research into transparency in model decisions, adversarial robustness and human-in-the-loop verification is recommended. Given the subjective nature of VS, annotated data can be both scarce and biased. We believe that combining pre-training with domain transfer is a promising direction. Furthermore, we see potential in developing reference-free evaluation frameworks—such as those based on reinforcement learning—to reduce reliance on annotated data.

**Future exploration of multimodality.** Other modalities (Zeng et al., 2017; See et al., 2017; Hermann et al., 2015) should be considered in future. At this point, we can apply ViSP objective to the video stream and the corresponding information-compression objectives to the audio and subtitle streams, formulating a multi-task problem. Notably, without ground-truth labels, learning fine-grained cross-modal collaboration remains challenging. We believe that acquiring the fundamental ability of information compression is simpler and more practical for summarization pre-training.

