# OpenReview forum: "Video Summarization Pretraining with Self-Discovery of Informative Frames"
_ICLR.cc/2026/Conference — ICLR 2026 Conference Withdrawn Submission_

### Official Review · Reviewer_XFbV · 2025-10-21

**Soundness:** 3
**Presentation:** 4
**Contribution:** 3
**Rating:** 4
**Confidence:** 2

**Summary:**

This paper addresses the limitation of existing video summarization (VS) pretraining methods—their heavy reliance on fixed pseudo-summaries that enforce a single perspective, failing to capture the subjective, diverse nature of valid summaries and hindering generalization. To resolve this, this paper proposes ViSP, a pseudo-summary-free pretraining framework that enables VS models to learn multifaceted frame importance from unlabeled videos. This approeach uses mutual information (MI) to measure the information overlap between sampled summaries and the original video, leverages contrastive learning with InfoNCE loss to approximate MI maximization, and adopts concrete-relaxation reparameterization to handle discrete summary sampling for gradient flow, supplemented by regularization terms for compactness and a binarization penalty to avoid trivial solutions

**Strengths:**

ViSP effectively targets a well-documented gap in VS pretraining by eliminating fixed pseudo-summaries, aligning its design with the ill-posed, subjective nature of VS and addressing the generalization conflict between prior pretraining methods and downstream tasks

The framework is conceptually simple and modular, with its core components decoupled from the base summarizer’s architecture, making it easy to integrate with existing models.

This paper provide a detailed ablation studies to validate the efficacy of key components.

**Weaknesses:**

Despite claiming ViSP works for “any neural video summarizer,” the paper exclusively uses CNN-based model for experiments. No tests are conducted on other SOTA architectures, such as Transformer-based models or multimodal (e.g., A2Summ). This raises doubts about ViSP’s compatibility with models that rely on sequential memory  or cross-modal cues. Additionally, frame features are extracted using a frozen GoogleNet, while modern VS models often use task-specific encoders

Experiments are confined to two small, legacy datasets (SumMe: 25 videos, TVSum: 50 videos) that lack diversity. Other newer, larger datasets (e.g., Daliy, CNN) are ignored, limiting ViSP’s validity for practical applications.

To validate ViSP’s advantage over pseudo-summary-based methods, the paper constructs “high-quality pseudo-summaries” using ground truth labels (e.g., top 15% frames from TVSum/SumMe) for ablation. This is unrealistic for real-world pretraining, where ground truth is unavailable.

**Questions:**

Can authors test ViSP on non-CNN architectures (e.g., Transformer-based models, GNN-based models) to validate its claim of working for “any neural video summarizer”

It would be interesting to test ViSP on newer/larger datasets to improve its practical validity beyond SumMe/TVSum?

Will you supplement F1 score (or other traditional metrics) to enable direct comparisons with legacy VS baselines?

Please add more case studies or failure cases analysis to better demonstrate ViSP’s efficacy across diverse video scenarios?

---

> ### Author Response · Authors · 2025-11-16
>
> Thank you very much for the detailed and thoughtful review. Below, we provide point-by-point responses to your questions.
>
> ### **Q1,Q2, W1, W2: Model Generality and Long Videos**
> >To find the results, we kindly refer the reviewer to the official comment we posted on the top. Due to the difficulty in annotation, newer and larger datasets for generic video summarization are scarce, so the utilization of unlabeled video is timely considered.
> We have carefully checked the datasets, but large text summarization datasets (CNN-Daily [1,2]) are not applicable to generic video summarization. However, we will discuss future directions in a new section and cite these datasets.
> >
> > - [1] Get To The Point: Summarization with Pointer-Generator Networks.
> > - [2] Teaching Machines to Read and Comprehend.
>
> ### **Q3: F1 Metrics**
> >Below we supplement the F1 score for reference.
> | Method | SumMe-F1 | TVSum-F1 |
> |---|---|---|
> | CSTA | 0.537±0.005 | 0.557±0.002 |
> | CSTA+ViSP  | 0.570±0.010 | 0.574±0.004 |
> >
> >ViSP has also achieved improvement in the F1 score, which demonstrates its effectiveness.
> >Below we provide a complete academic background. Before the work [1] analyzed the limitations of F1, video summarizers were mainly optimized by calculating cross-entropy loss based on the binary chosen labels of frames. After the work [1], using MSE loss to fit the evaluation scores, and finally evaluating the correlation coefficients after KTS + knapsack processing are commonly used.
> >
> > - [1] Rethinking the Evaluation of Video Summaries
>
> ### **Q4: More Case Studies**
> >We have provided successful cases in Figure 8. We will find some failure cases and add them in the Appendices of the paper.
>
> ### **W3: Advantage over Pseudo-Summary-based Methods**
> >We construct high-quality pseudo-summaries using ground truth labels only for ablation baseline, not for our main results. This is to illustrate that even with high-quality pseudo-summaries, the transferability will decline due to the introduction of the fixed view problem, compared to our view-free pretraining. We also highlight this point in Section 4.4.1.

---

### Official Review · Reviewer_2dHv · 2025-10-27

**Soundness:** 3
**Presentation:** 2
**Contribution:** 3
**Rating:** 4
**Confidence:** 5

**Summary:**

The authors propose ViSP, a pretraining method that avoids pseudo-summaries and teaches models to identify meaningful summaries from unlabeled videos. This approach allows fine-tuning to adapt to specific downstream tasks, improving supervised learning. ViSP is simple, versatile, and empirically effective, achieving state-of-the-art performance on benchmark datasets SumMe and TVSum.

**Strengths:**

1. The authors propose a novel pretraining framework designed to learn versatile frame importance by examining diverse summaries within each unlabeled video.
2. They develop the ViSP foundation summarizer, which distinguishes and explores more representative summaries using mutual information estimation and learning-based sampling.
3. Their results show that ViSP effectively enhances the performance of state-of-the-art video summarizers.

**Weaknesses:**

1. The work does not provide sufficient evidence of generalization. Experiments are conducted only on a single model (CTSA), leaving it unclear whether the proposed method would yield consistent gains when applied to other summarization architectures.
2. It is not clearly justified how the proposed framework enables the model to capture multifaceted frame importance, given that the model is still fine-tuned using ground-truth summaries.
3. The method adopts CTSA as the video encoder during pretraining. To obtain more comprehensive video and summary representations, more advanced video encoders should be considered instead of reusing the same encoder as the downstream summarizer.
4. The authors claim that the framework learns “versatile” frame importance, but the manuscript does not provide convincing evidence for this claim, nor does it clearly articulate the benefit of such versatility.
5. The evaluation relies on small and outdated datasets. Validation on larger and more recent benchmarks is necessary to establish the method’s effectiveness.

**Questions:**

Please refer to the Weaknesses section.

---

> ### Author Response · Authors · 2025-11-16
>
> Thank you very much for the thoughtful and constructive reviews. Below, we provide point-by-point responses to your questions.
>
> ### **W1: Transferability of ViSP across Model Architectures**
> >To find the transferability results, we kindly refer the reviewer to the official comment we posted on the top.
>
> ### **W2: Supplementary Results on Diversity**
> >In lines 72-82, the paper argues that forging diversity in pre-training is beneficial to supervised fine-tuning, so it is necessary to use real summaries for fine-tuning. We kindly refer to the official comment; by increasing the temperature parameter λ, the diversity presented in the entropy of per-frame sampling probabilities increases.
>
> ### **W3: Input Feature of Video Encoder Should be the Same as that of the Summarizer**
> >ViSP is a relaxed implementation of the objective function (Eq. 1), grounded in an information-bottleneck (line 172 and Appendices 4.3). This indicates that $X_s$ is searched in the feature space defined based on $X_o$.
> >Changing the feature space where the video encoder is located, the perception of information and the search for summaries occur in different spaces, and the pre-trained reward and fine-tuned reward are also based on different feature spaces respectively, resulting in deviation.
> >Below, we report the results of ViSP+CSTA using videoprism-public-v1-base [1] as video encoder during pretraining.
> | Video Encoder | SumMe-τ | SumMe-ρ | TVSum-τ | TVSum-ρ | Avg. |
> |---|---|---|---|---|---|
> | CSTA | 0.273 | 0.305 | 0.201 | 0.263 | 0.260 |
> | VideoPrism  | 0.246 | 0.275 | 0.197 | 0.259 | 0.244 |
> >
> >We found that the mutual information estimation performs better when the features used by the video encoder and the summarizer are consistent. Intuitively, even if the video encoder uses richer information to improve the mutual information estimation, the summarizer cannot utilize this additional information for summarization, resulting in a gap.
> >- [1] VideoPrism: A Foundational Visual Encoder for Video Understanding
>
> > However, using the same input features as the summarizer but replacing it with a Transformer (depth 6, 8 heads, hidden dimension of 512) yields results (**0.258 in Avg.**) close to those of the main experiment CSTA. This indicates that the readout module of video feature can be changed, but the input feature should be consistent.
>
> ### **W4: Clarification on Versatility**
> >Versatility is demonstrated by the fact that one pretrained ViSP checkpoint, when fine-tuned separately on TVSum and SumMe, results in summarizers with better performance. This indicates that using unlabeled video data can create a shared foundation model for different datasets.
>
> ### **W5: Results on Large Videos**
> >We kindly refer the reviewer to the official comment on the top, where we supplemented the experiment on the hour-long video (QFVS). TVSum and SumMe are standard and widely adopted benchmarks for generic video summarization. Due to the difficulty in annotation, labeled video summarization data is very scarce, so the utilization of unlabeled video is timely considered.

---

### Official Review · Reviewer_2pnU · 2025-10-31

**Soundness:** 3
**Presentation:** 3
**Contribution:** 3
**Rating:** 6
**Confidence:** 4

**Summary:**

- This paper introduces a representation learning framework for video summarization that enhances the quality of summary clip features by maximizing their mutual information with the original video features. The approach aims to capture diverse frame importance without relying on pseudo-summaries produced by existing pretrained models. Specifically, the method employs an InfoNCE loss to maximize a lower bound on the mutual information between the sampled summary and the source video, treating them as a positive pair. To enable differentiable frame selection, the approach leverages reparameterization techniques such as Gumbel-Softmax and Concrete relaxation, allowing for a softened binary mask. The model further regularizes summary length and temporal smoothness during training to control the budget and mitigate issues of sparsity or fragmentation. Evaluation focuses on rank-based metrics (Kendall/Spearman), using a fixed summary pipeline (KTS segmentation and knapsack under a 15% budget). Experiments on SumMe and TVSum demonstrate consistent improvements over strong baselines, highlighting the framework’s practical effectiveness for learning frame importance without reliance on fixed pseudo-summaries.

**Strengths:**

- The paper is clearly written and technically accessible, with a transparent connection between the method and its implementation, particularly in how frame scores and video-level representations are incorporated into the contrastive objective.
- The evaluation protocol is well-justified for the task, employing rank-based metrics and a fixed summary construction pipeline, which enhances the comparability of results.
- The approach is practical, which adopts differentiable selection via Gumbel-Softmax/Concrete relaxation, combined with lightweight deployment since the pretraining encoder can be omitted at inference, which facilitates straightforward integration with existing video summarization pipelines.

**Weaknesses:**

- The paper argues that sampling diverse summaries during pretraining is beneficial. However, the empirical evidence primarily compares "diverse sampling" to "fixed summaries," without directly analyzing between-sample diversity for the same video. A more convincing analysis would include metrics such as the distribution of pairwise IoU/Jaccard indices across multiple independently sampled summaries for each video, or the entropy/effective support size of the per-frame selection probabilities.
- While ranking correlation metrics capture the quality of frame importance ordering, they do not necessarily reflect the quality of final summaries produced after KTS and knapsack selection. The robustness of the reported gains to alternative segmentation strategies (e.g., uniform segments or VSUMM) or to omitting the knapsack step is not demonstrated, leaving a gap between improvements in ranking and actual end-to-end summary quality.
- The “unseen” evaluation setting, which leverages partial in-domain data, is reasonable; however, the process for sampling and de-duplication against test videos, including near-duplicate filtering, should be described more rigorously to ensure the integrity of the evaluation.

**Questions:**

- For each video, could you sample k≥5 summaries and report the distribution of pairwise IoU, along with the entropy or effective support size of the per-frame selection probabilities? This would more directly support the claim of capturing “diverse perspectives” beyond coverage with a single reference.
- If sweeping the temperature or regularization parameter to modulate diversity, do you observe a monotonic relationship between the chosen diversity metric and Kendall/Spearman correlation? Even a limited sweep could help clarify causality.
- Do results hold when using a different shot segmenter or omitting the knapsack step? Reporting rank-biased overlap (RBO) or basic coverage/diversity metrics on the final summaries would better bridge the gap between improved rankings and true end-to-end summary quality.
- How are in-domain videos sampled for the “unseen” pretraining pool, and what steps are taken to prevent leakage of test videos or near-duplicates into pretraining? For out-of-domain scaling, a brief analysis of domain distance (e.g., using CLIP-space statistics) versus performance gains would further strengthen the conclusions.

**Details Of Ethics Concerns:**

No ethics concerns.

---

> ### Author Response · Authors · 2025-11-16
>
> We sincerely appreciate your recognition and constructive feedback. Below, we provide point-by-point responses to your questions.
>
> ### **Q1,Q2,W1: The Relationship Between Temperature, Pretraining Diversity, and FineTuning Performance**
> >To find the results and analyze, we kindly refer the reviewer to the official comment we posted at the top.
> >We found that the pairwise IoU computed from hard-sampled summaries depends on an ill-defined summarization size, making it difficult to analyze reliably. Therefore, in our official comment at the top, we focus primarily on the entropy of the per-frame selection probabilities, which provides a more stable and interpretable signal. For completeness, we also compute the pairwise IoU over five sampled summaries per video and report the averaged results below as a reference for some summarization size.
> | Size | λ=0.005 | λ=0.01 | λ=0.05 | λ=0.5 | λ=1.0 | λ=5.0 |
> |-----|------|-------|-------|-------|-------|-------|
> | 10% | 0.1107±0.0128 | 0.1085±0.0132 | 0.1079±0.0143 | 0.0818±0.0122 | 0.0692±0.0115 | 0.0522±0.0099 |
> | 30% | 0.4298±0.0363 | 0.4244±0.0330 | 0.4137±0.0420 | 0.2849±0.0150 | 0.2335±0.0150 | 0.1802±0.0100 |
> | 60% | 0.8233±0.0559 | 0.8746±0.0559 | 0.9687±0.0118 | 0.6691±0.0237 | 0.5406±0.0164 | 0.4343±0.0081 |
> | 90% | 0.8376±0.0055 | 0.8406±0.0057 | 0.8767±0.0094 | 0.9401±0.0089 | 0.8921±0.0099 | 0.8191±0.0039 |
> >
> >Due to randomness of this metric, when the summarization size is large, the pairwise IoU does not strictly decrease monotonically; however, when the summarization size is small, the monotonic decrease is obvious.
>
> ### **Q3,W2: Performance gain without KTS+knapsack**
> >We agree that conducting an ablation study on KTS+knapsack can further strengthen the conclusions. In particular, KTS and knapsack selection are used to generate shot-level scores, allowing the summarizer to be evaluated on the shot-level labels provided by TVSum.
> In fact, skipping KTS and knapsack is equivalent to evaluating frame-level relevance. The frame-level correlation coefficient without KTS+knapsack is as follows.
> | w/o KTS+knapsack | SumMe-τ | SumMe-ρ | TVSum-τ | TVSum-ρ | Avg. |
> |---|---|---|---|---|---|
> | CSTA | 0.133±0.013 | 0.181±0.017 | 0.334±0.006 | 0.479±0.006 | 0.282±0.008 |
> | CSTA+ViSP | 0.163±0.002 | 0.219±0.002 | 0.342±0.006 | 0.487±0.009 | 0.303±0.004 |
>
> ### **Q4,W3: Supplementary Results on Out-of-Domain Scaling**
> >For the experiment in Fig. 7, after we initially collected and constructed out-of-domain (OOD) data, we filtered in-domain videos based on video duration and titles, and randomly split them. We designed new experiments based on the reviewer's feedback as follows.
> >To further strengthen the conclusion, below we supplement the domain distance versus performance gains. We calculate the L2 distance between the average frame feature of each out-of-domain video and the average frame feature of the nearest neighbor in-domain video, then sort all out-of-domain videos, and divide them equally into three sets for pre-training.
> | Metric | OOD split-1 | OOD split-2 | OOD split-3 |
> |-----|-------|-------|-------|
> | nearest in-domain distance | 0.314 | 0.537 | 0.608 |
> | Avg. performance  | 0.252 | 0.245 | 0.238 |
> >
> >The nearest in-domain distance is calculated based on the two closest data points between the OOD split and the in-domain split.
>
> > There is no need to specifically remove near-duplicate videos because we can quantify distribution shifts based on the nearest in-domain distance. In-domain data is the unlabeled training set in TVSum, SumMe benchmarks.

---

### Official Review · Reviewer_c7Xx · 2025-10-31

**Soundness:** 2
**Presentation:** 3
**Contribution:** 2
**Rating:** 4
**Confidence:** 4

**Summary:**

The paper proposes an annotation-free pretraining framework for video summarization. It evaluates the frame importance without relying on pseudo summaries. It maximizes the mutual information between the original video and its sampled summary through contrastive learning, and uses reparameterization sampling for differentiable frame selection. Authors conduct some interesting experiments on SumMe and TVSum to demonstrate the performance improvements.

**Strengths:**

1. It introduces a well-motivated pseudo-summary-free pretraining framework that learns informative frame representations by maximizing mutual information.
2. The theoretical foundation is clearly established through the use of a contrastive lower bound on mutual information.
3. Experimental results are mostly promising by consistent gains over baseline methods.

**Weaknesses:**

1. Some of the mathematical formulations, particularly the sampling and optimization sections, are presented too tersely. The role of auxiliary variable Z is not rigorously derived.

2. Experimental evaluation is limited to relatively small datasets (SumMe and TVSum), and does not test the model on larger data set, such as VTW (2529 videos) or multimodal benchmarks. This restriction limits confidence in generalization capability to more complex or real-world scenarios.

**Questions:**

1. For the auxiliary variable Z and Eq. (9): under what data distributions does marginalization with Z lead to a better approximation? Can its influence on variance estimation and gradient stability be quantified, for example by providing an error bound or empirical comparison?

2. Regarding model generality: beyond CSTA, how does ViSP perform when transferred to other summarization architectures? Does it still yield statistically significant improvements across architectures?

3. For cross-domain and long videos (over 30 minutes) or high-motion scenes (e.g., sports or documentaries), how does the method behave and where does it fail? Would hierarchical sampling or temporal segmentation be necessary to stabilize the InfoNCE objective?

4. In a multimodal setting, if audio or subtitle streams are introduced as additional modalities, how should ViSP’s mutual information estimation be reformulated—would a multi-view InfoNCE objective be appropriate to mitigate single-view bias? This question should be considered in the future work.

---

> ### Author Response · Authors · 2025-11-16
>
> Thank you very much for the insightful review. Below, we provide point-by-point responses to your questions.
>
> ### **Q1,W1: Deeper Analysis of Z and Eq. (9)**
> > For the convenience of discussion, we consider the mask formulation of each frame.
> >- Multiplicative masking (i.e., gating): $\hat{x}=x\cdot {y}$
> >- Marginalization-based masking (Eq. (9)): $\hat{x}=x\cdot {y}+(1-{y})\cdot z$
> >
> > for each frame $x \in X_o$, we have corresponding frame-level mask value $y \in \hat{Y}$ and auxiliary variable $z\in Z$.
> >Assume frame features corresponding to different semantics follow Gaussian distributions. For two distinct frames $x_1 \sim q_1(x_1)$, $q_1(x_1)=\mathcal{N}(\mu_1,\sigma_1^2)$ and $x_2 \sim q_2(x_2)$, $q_2(x_2)=\mathcal{N}(\mu_2,\sigma_2^2)$, gating with a small $y$ yields $\hat{x}_1 \sim \mathcal{N}(y\mu_1, y^2\sigma_1^2)=q_1(y\cdot x_1)$ and $\hat{x}_2 \sim \mathcal{N}(y\mu_2, y^2\sigma_2^2)=q_2(y\cdot x_2)$, which still preserve the same semantic distinctions when $y\neq0$ because $KL(q_1(x_1)||q_2(x_2))=KL(q_1(y \cdot x_1)||q_2(y \cdot x_2))$ for Gaussian distributions.
> Eq. (9) introduces an auxiliary variable $z\sim p_z$ specifically to eliminate this residual discriminability by reducing the distance between distributions $p(\hat{x}_1)$ and $p(\hat{x}_2)$ when $y\to 0$.
>
> >**Theorem.**
> >>Let $x_1\sim q_1,; x_2\sim q_2$ and fix $0<y<1$. Let $z$ (and $z'$) have law $p_z$. Define:
> >>
> >>$\qquad v = yx_1 + (1-y)z,\qquad w = yx_2 + (1-y)z'$
> >>
> >>Let $p_{y x_i}$ denote the law of $y x_i$ and $p_v,p_w$ the laws of $v,w$. Then
> >>
> >>$\qquad KL\big(p_v|p_w\big)\le KL\big(p_{y x_1}|p_{y x_2}\big)$
> >
> >**Proof.** (We view the construction of $v,w$ as two successive Markov kernels.)
> >> Deterministic scaling $S:x\mapsto yx$. For $i=1,2$ write $q_i$ for the law of $x_i$. Then the law after scaling is $p_{y x_i} = q_i S$.
> >
> >>Additive-noise is kernel $K$: Given $s$ output $s+(1-y)z$. This kernel $K(\cdot\mid s)$ is the same for both inputs. Applying $K$ to $p_{y x_i}$ yields $p_v = (q_1 S)K$ and $p_w=(q_2 S)K$.
> >
> >> Use the data-processing property of $KL$, for any two input distributions $(P,Q)$ and any Markov kernel $K$, $KL(PK|QK)\le KL(P|Q)$. Apply this with $P=q_1S$ and $Q=q_2S$ and kernel $K$, we have $KL\big((q_1S)K\ \big|\ (q_2S)K\big)\le KL(q_1S|q_2S)$
> >
> > If the noise kernel $K$ is nontrivial and $p_{y x_1}\neq p_{y x_2}$, the inequality is typically strict: adding noise via the same channel reduces distinguishability.
>
> >When $y\to 1$, $\hat{x}\to x$; when $y\to 0$, $\hat{x}\to z \sim p_z$, removing semantic cues. In Section 4.3.1 we find that naively using real frame features as $z$ reduces pretraining benefits. We attribute this to: (i) the backbone features are not parameterized by challenge distribution thus gating-based mask can easily handle; (ii) directly using features from other videos as $z$ keeps them in the same distribution as unmasked frames, which may confuse the encoder. To fairly verify the utility of Eq. (9), we adopt controlled setting with $h\sim\mathcal{N}(x,1)$ as frame features and $z\sim\mathcal{N}(0,1)$. Experiments show that Eq. (9) consistently outperforms simple multiplicative masking under this controlled setting.
> | Method | SumMe-τ | SumMe-ρ | TVSum-τ | TVSum-ρ | Avg. |
> |---|---|---|---|---|---|
> | CSTA      | 0.203±0.004 | 0.227±0.004 | 0.087±0.003 | 0.114±0.003 | 0.158±0.003 |
> | CSTA+ViSP | 0.231±0.003 | 0.258±0.004 | 0.074±0.010 | 0.098±0.014 | 0.165±0.007 |
> | w/ Eq(9)  | 0.233±0.011 | 0.259±0.013 | 0.086±0.002 | 0.114±0.002 | 0.173±0.005 |
> >
> >We can see similar parameterization of image features using normal distribution in VAE.
> >However, currently advanced summarizers are not based on such features, so we adopted the simplest adaptation as a reference to support the extra analysis.
>
> ### **Q2,Q3: Model Generality and Long Videos**
> >To find the results, we kindly refer the reviewer to the official comment we posted on the top. Developing a summarizer that processes long videos is another research topic, where it is not a capability claimed by ViSP. Therefore, we consider the simplest adaptation for long-video parsing to obtain a referable result.
>
> ### **Q4,W2: Future Exploration of Multimodality**
> >We agree that unlabeled multimodal data should be considered in future work. At this point, we can apply the ViSP objective to the video stream and the corresponding information-compression objectives to the audio and subtitle streams, formulating the setting as a multi-task problem. Notably, without ground-truth labels, learning fine-grained cross-modal collaboration remains highly challenging. We believe that acquiring the fundamental ability of information compression is simpler and more practical for summarization pre-training.
> >
> > As our work focuses on generic video summarization, we will discuss future directions and cite these multimodal datasets, e.g., video-QA dataset (VTW [1]).
> > - [1] Leveraging Video Descriptions to Learn Video Question Answering

---

### Author Response · Authors · 2025-11-15
**Major Concerns Across All Reviews**

We sincerely thank all reviewers and the area chair for their time, effort and thoughtful feedback.

After carefully synthesizing the reviews, we address the major/common concerns below. Individual, reviewer-specific comments will be provided in the respective review threads.

### **Transferability of ViSP across Model Architectures**
>In Table 2 of paper, we have provided the results for CSTA+ViSP as well as GoogleNet+ViSP.
Below, we supplement the results of combining with more model architectures.
| Model | SumMe-τ | SumMe-ρ | TVSum-τ | TVSum-ρ | Avg. |
|---|---|---|---|---|---|
| CSTA | 0.246 | 0.274 | 0.194 | 0.255 | 0.242 |
| CSTA+ViSP | 0.273 | 0.305 | 0.201 | 0.263 | 0.260 |
| GoogleNet | 0.176 | 0.197 | 0.129 | 0.163 | 0.166 |
| GoogleNet+ViSP | 0.198 | 0.220 | 0.131 | 0.166 | 0.179 |
| Transformer | 0.184 | 0.205 | 0.087 | 0.106 | 0.146 |
| Transformer+ViSP | 0.209 | 0.234 | 0.148 | 0.195 | 0.196 |
| GNN | 0.175 | 0.195 | 0.059 | 0.067 | 0.124 |
| GNN+ViSP | 0.187 | 0.208 | 0.128 | 0.169 | 0.173 |
>
>For Transformer, we use the standard transformer provided by pytorch.nn, with a depth of 6, 8 num heads, a hidden layer dimension of 512; for GNN, we generate edges for adjacent frames and use GCNConv provided by torch-geometric to construct the GNN, with a depth of 5, a hidden layer dimension of 2048; dropout is set to 0.1, and finally there is an additional linear layer to project each hidden feature to 1 to generate the frame score.

>We reached the same conclusion for the presented architectures, with p-values<0.05 for all improvements.

### **The Relationship Between Temperature, Pretraining Diversity, and FineTuning Performance**
>We agree that the entropy of the per-frame selection probabilities (EPFSP) is very practical, and the variation of the entropy of the per-frame selection probabilities with temperature λ is as follows. Average performance on SumMe and TVSum is also reported.
| Metric | λ=0.005 | λ=0.01 | λ=0.05 | λ=0.5 | λ=1.0 | λ=5.0 |
|-----|------|-------|-------|-------|-------|-------|
| EPFSP | 0.004 | 0.008 | 0.042 | 0.339 | 0.501 | 0.677 |
|  Avg. Performance | 0.239 | 0.244 | 0.256 | 0.260 | 0.249 | 0.234 |
>
>The pretraining configuration is the same as the main experiment in the paper (Table 2).
We found that as λ increases, the entropy of the per-frame selection probabilities (i.e., diversity) increases. However, there is no monotonic relationship between the diversity metric and Kendall/Spearman correlations and there is a tradeoff. Excessively high diversity means that redundant or noisy frames are difficult to distinguish. Excessively low diversity causes the pretrained distribution to fail to cover potential perspectives, similar to another type of fixed pseudo-summary.

>These conclusions are consistent with Fig. 1, lines 72-82 and Section 4.4.2 of the paper. Pretraining aims to remove more redundancy while retaining useful information to cover as many potential perspectives as possible; there is a tradeoff. In this way, finetuning only requires selecting perspectives, making it easier to learn.

### **Generalization to Large/Long Videos**
>QFVS [1] (Sharghi et al., 2017) provides hour-long videos (Appendices.4). Running KTS for QFVS evaluation takes 4,000 hours (OOT) for each video, and the number of frames in a video also exceeds the processing capacity of the summarizer. Therefore, we have made some modifications to obtain referable results. We extract 5,000 frames from each video to form a new original video that supports CSTA's context window. On this new video, we perform video summarization pretraining and finetuning. Finally, we combine the importance scores generated separately for each segment through maximum pooling and directly calculate the correlation coefficients without using KTS+knapsack (OOT) for shot selection.
| Model | CSTA | CSTA+ViSP pretrained on TVSum+SumMe | CSTA+ViSP pretrained on TVSum+SumMe+QFVS |
|---|---|---|---|
| QFVS-τ |0.030±0.002|0.044±0.011|0.084±0.014|
| QFVS-ρ |0.036±0.002|0.050±0.012|0.075±0.014|
>
>The results of the leave-one-out experiment on QFVS show that ViSP can boost CSTA without using QFVS for pretraining, and bring a 2-fold performance gain when using unlabeled QFVS data.

>ViSP aims to benefit the base summarizer by using unlabeled video data. Despite adopting the simplest scheme, the experimental results show that ViSP can also improve the CSTA on long videos. Hierarchical sampling, temporal segmentation, and the state machine for long video summarization are worthy of further exploration in other papers.

>- [1] Query-Focused Long Video Summarization

---

### Author Response · Authors · 2025-11-25
**PDF Update (Nov 24) — Added Supplementary Results Following Reviewer's Suggestion**

Dear Reviewers and AC,

Following suggestions, we updated the main paper and Appendices on Nov 24. The newly added results and additional case studies in the revision are highlighted in red.

The update adds more analytical experiments and does not conflict with any original conclusions.

Best,
Authors

---

### Note · Authors · 2026-01-28

I have read and agree with the venue's withdrawal policy on behalf of myself and my co-authors.

---

### Meta-Review · Area_Chair_d1Tb · 2026-01-15

**Summary:**

This submission proposes ViSP, a pseudo-summary-free pretraining framework for video summarization that maximizes mutual information between a sampled summary and the full video via an InfoNCE-style objective, using differentiable sampling with regularizers. Reviewers broadly found the idea well-motivated and the empirical gains consistent on SumMe/TVSum, but raised recurring concerns about limited evaluation breadth (small/legacy datasets), generalization beyond the main architecture/feature pipeline, and clarity/rigor of some technical details (notably the auxiliary variable Z / Eq. (9) and the diversity claim). Additional concerns included whether improvements translate to end-to-end summary quality under alternative selection pipelines, and whether the “versatility” claim is sufficiently evidenced.

**Reviewer Concerns:**

The rebuttal and the most recent update addressed several key concerns with additional experiments and clarifications: (i) transferability across multiple model architectures (including Transformer/GNN variants) with statistically significant gains, (ii) explicit analysis of diversity vs. temperature (EPFSP sweep) and the non-monotonic tradeoff between diversity and downstream performance, (iii) a long-video/generalization experiment on QFVS with a reasonable adaptation, and (iv) ablations bridging ranking improvements to evaluation pipeline choices (including results without KTS+knapsack), plus added explanation and ablation regarding Z/Eq. (9). Outstanding concerns are primarily about external validity at scale—evaluation remains centered on SumMe/TVSum with only adapted evidence for long videos, and the paper still lacks convincing validation on larger, modern, diverse benchmarks or truly multimodal settings; additionally, while the
Z/Eq. (9) explanation is improved, some reviewers may still view the theory as insufficiently rigorous/quantified (e.g., no clear error bounds/variance-stability analysis), and the “versatile/multifaceted importance” claim may still feel stronger than what the evidence fully supports.

**Reviewer Scores:**

With full discussion, score changes would likely be limited and insufficient to reach consensus for acceptance. Reviewer 2pnU would likely remain at 6. While several analytical concerns were addressed, the core limitation on dataset scale and end-to-end validation persists. Reviewers c7Xx, 2dHv, and XFbV would likely move only marginally from 4 to 4–5, acknowledging improved clarity, additional architectures, and long-video experiments. The remaining concerns are on the external validity, theoretical rigor of the auxiliary variable formulation, and the strength of evidence supporting the “versatility” claim. Overall, the post-discussion score distribution would remain mixed and slightly below the acceptance threshold.

---

### Decision · Program_Chairs · 2026-01-26

Reject